# Blackcurrant Improves Diabetic Cardiovascular Dysfunction by Reducing Inflammatory Cytokines in Type 2 Diabetes Mellitus Mice

**DOI:** 10.3390/nu13114177

**Published:** 2021-11-22

**Authors:** Hye-Yoom Kim, Jung-Joo Yoon, Hyeon-Kyoung Lee, Ai-Lin Tai, Yun-Jung Lee, Dae-Sung Kim, Dae-Gill Kang, Ho-Sub Lee

**Affiliations:** 1Hanbang Cardio-Renal Research Center, Professional Graduate School of Oriental Medicine, Wonkwang University, Iksan 54538, Korea; hyeyoomc@naver.com (H.-Y.K.); mora16@naver.com (J.-J.Y.); gorud0170@naver.com (H.-K.L.); 775452499@gg.com (A.-L.T.); shrons@wku.ac.kr (Y.-J.L.); 2College of Oriental Medicine, Professional Graduate School of Oriental Medicine, Wonkwang University, Iksan 54538, Korea; 3Hanpoong Pharm and Foods Co., Ltd., Wanju 55316, Korea; kimezz@naver.com

**Keywords:** blackcurrant, type 2 diabetes mellitus, diabetic cardiomyopathy, cardiovascular, inflammation

## Abstract

Diabetic cardiovascular dysfunction is a representative complication of diabetes. Inflammation associated with the onset and exacerbation of type 2 diabetes mellitus (T2DM) is an essential factor in the pathogenesis of diabetic cardiovascular complications. Diabetes-induced myocardial dysfunction is characterized by myocardial fibrosis, which includes structural heart changes, myocardial cell death, and extracellular matrix protein accumulation. The mice groups in this study were divided as follows: Cont, control (db/m mice); T2DM, type 2 diabetes mellitus mice (db/db mice); Vil.G, db/db + vildagliptin 50 mg/kg/day, positive control, dipeptidyl peptidase-4 (DPP-4) inhibitor; Bla.C, db/db + blackcurrant 200 mg/kg/day. In this study, Bla.C treatment significantly improved the homeostatic model evaluation of glucose, insulin, and insulin resistance (HOMA-IR) indices and diabetic blood markers such as HbA1c in T2DM mice. In addition, Bla.C improved cardiac function markers and cardiac thickening through echocardiography. Bla.C reduced the expression of fibrosis biomarkers, elastin and type IV collagen, in the left ventricle of a diabetic cardiopathy model. Bla.C also inhibited TD2M-induced elevated levels of inflammatory cytokines in cardiac tissue (IL-6, IL-1β, TNF-α, and TGF-β). Thus, Bla.C significantly improved cardiac inflammation and cardiovascular fibrosis and dysfunction by blocking inflammatory cytokine activation signals. This showed that Bla.C treatment could ameliorate diabetes-induced cardiovascular complications in T2DM mice. These results provide evidence that Bla.C extract has a significant effect on the prevention of cardiovascular fibrosis, inflammation, and consequent diabetes-induced cardiovascular complications, directly or indirectly, by improving blood glucose profile.

## 1. Introduction

Type 2 diabetes mellitus (T2DM) is known to cause metabolic disorders such as obesity, insulin resistance, and liver dysfunction and causes dysfunction of major organs [1]. Representatively, the major pathophysiological mechanism linking type 2 diabetes to cardiovascular risk is diabetic dyslipidemia characterized by low-density lipoprotein cholesterol (LDL-C) and triglyceride plasma levels [2]. In addition, T2DM is known to induce diabetic cardiomyopathy as a risk factor for heart failure, and cardiovascular disease is one of the leading causes of death in diabetic patients [3]. Death from diabetic cardiovascular complications is so dangerous that it accounts for more than 80% of all diabetes deaths [4]. In addition, T2DM-induced inflammation is known to increase the incidence of cardiovascular dysfunction and increase mortality [5]. Therefore, various studies are being conducted to find new therapeutic targets to delay diabetes or reduce mortality due to increased diabetic complications [6]. Cardiovascular disease is one of the most common complications of type 2 diabetes. Abnormal changes in cardiovascular function due to abnormal blood pressure significantly alter hemodynamic stress [7]. Inflammation plays a pivotal role in the pathogenesis of diabetic complications in both microvascular and cardiovascular systems [8]. Metabolic abnormalities in diabetes cause inflammation not only in the myocardium, but also in the endothelial cells of large and small blood vessels [9]. Inflammation is an important factor in chronic diseases including obesity, cardiac disease, vascular disease, and metabolic syndrome in pathophysiology [10]. T2DM increases insulin resistance, and the inflammatory cytokine interleukin-6 (IL-6) increases plasma triglycerides, resulting in dyslipidemia [11]. Interleukins reduce high-density lipoprotein cholesterol (HDL-C) and triglyceride levels, leading to structural and functional changes in the cardiovascular system [12]. Ingestion of functional foods with anti-inflammatory effects is known to have the effect of extending the lifespan of animal models by reducing lipid accumulation in the liver, improving immune response and antioxidant activity, suppressing proinflammatory cytokines, and regulating energy balance [13,14]. Therefore, we studied the amelioration effect of blackcurrant in diabetic cardiomyopathy.

Blackcurrant contains vitamins, carotenoids, and flavonoids [15], and several studies have reported that it has antiulcer, anticonvulsant, and protective effects against degenerative diseases [16] and antioxidant effects [17]. Anthocyanins, which are abundantly contained in berries such as blackcurrant, remove oxidative stress via an Nrf2-mediated antioxidant mechanism [18]. Blackcurrants are also known to have anti-inflammatory and cytoprotective effects [19], and furthermore, blackcurrant is known to protect against kidney stones [20]. Although a study has reported the effects of berry consumption on metabolic parameters [21], the effect of blackcurrant on improving cardiovascular dysfunction in diabetic metabolic disorders has not been reported. Therefore, this study was designed to confirm the improvement effect of blackcurrant extract in a mouse model of type 2 diabetic cardiovascular dysfunction.

## 2. Materials and Methods

### 2.1. Blackcurrant Extraction

Blackcurrant was purchased from National Farm (Jeongeup, Korea), and 400 g of blackcurrant was boiled with 3 L of distilled water at 100 °C for 2 h. The resulting extract was filtered through a qualitative filter paper (Advantec No. 3, Tokyo, Japan) and centrifuged at 4 °C and 3000 rpm for 20 min. Then, the obtained supernatant was concentrated in a rotary vacuum evaporator (N-11, EYELA, New York, NY, USA). The extract was dried using a freeze dryer (FD8508, Ilshin Lab, Yangju, Korea) to obtain 64 g in powder form and stored at −70 °C. The extracted blackcurrant specimens were stored at the Cardio-renal Syndrome Research Center in Wonkwang University (No. HBF211–08).

### 2.2. High-Performance Liquid Chromatography Analysis

Quantitative analysis of blackcurrant was performed using high-performance liquid chromatography (HPLC) equipment (LC-2030C 3D, Shimadzu, Kyoto, Japan). Among the anthocyanin standards, the four compounds known to be contained in blackcurrant are as follows: delphinidin-3-*O*-rutinoside, delphinidin-3-*O*-glucoside, cyanidin-3-*O*-rutinoside, and cyanidin-3-*O*-glucoside (Figure 1A–D). Analysis of the four compounds was performed using a Capcell Pak C18 ACR column (250 × 4.6 mm; 5 μm; injection volume: 10 μL; run time: 60 min) (Shiseido, Tokyo, Japan).

### 2.3. Experimental Animals

Experiments were performed using male type 2 diabetes mellitus mice (C57BLKS/+Lepr^db^, db/db mice, 8 weeks) and control mice (C57BLKS/J, db/m mice, 8 weeks) purchased from Jackson Laboratory (Bar Harbor, ME, USA). The mice were bred in individual cages in a room maintained in a temperature- and humidity-controlled environment (light–dark cycle of 12–12 h). The mice groups in this study were divided as follows: Cont, control (db/m mice); T2DM, type 2 diabetes mellitus mice (db/db mice); Vil.G, db/db+vildagliptin 50 mg/kg/day, positive control, dipeptidyl peptidase-4 (DPP-4) inhibitor; Bla.C, db/db + blackcurrant 200 mg/kg/day. The concentration of blackcurrant used in this study was determined based on the results of previous studies [22]. Blackcurrant was administered daily in free drinking water for 10 weeks, and the control group was administered distilled water in the same manner. This study was approved by the IACUC (Institutional Animal Care and Use Committee) of Wonkwang University (WKU21–18).

### 2.4. Assessment of Insulin Resistance and Oral Glucose Tolerance Test

Oral glucose tolerance test (OGTT) was performed prior to the sacrifice of mice at 10 weeks of Bla.C administration. Briefly, blood samples were collected via tail vein after an overnight fast of 10–12 h prior to performing OGTT. After oral administration of glucose (2 g/kg) in mice each group (fasted state), a trace amount was collected from the tail vein at 30, 60, 90, and 120 min using a blood glucose meter to measure blood glucose (OneTouch Ultra, LifeScan, Milpitas, CA, USA). The levels of mouse insulin (AKRIN-011T, Fujifilm Value from Innovation, Tokyo, Japan) were determined using commercially available ELISA kits. The insulin resistance (homeostatic model assessment for insulin resistance, HOMA-IR) assessment was calculated with reference to existing literature formulas [23].

### 2.5. Echocardiographic Analysis

Before the echocardiography examination, mice were anesthetized in a chamber with isoflurane (1.5%) in mixed gas containing oxygen (95%) and carbon dioxide (5%), and chest fur was removed using a depilatory cream. Echocardiography was performed with the ultrasound imaging system VINNO 6 (frequency 23 MHz, Vinno Corporation, Suzhou, China). For echocardiography, 4 to 5 consecutive cardiac cycles were measured to obtain an average value, and M-mode images were captured and stored for analysis. The fractional shortening (FS), ejection fraction (EF), left ventricular end-systolic volume (LVESV), left ventricular end-diastolic volume (LVEDV), left ventricular mass (LVd Mass), left ventricular internal dimension in diastole (LVIDd), and left ventricular posterior wall thickness (LVPWd) were calculated.

### 2.6. Determination of Lipid Profile and Atherogenic Indexes

After completion of the experiment, mouse blood was collected in a blood collection tube containing heparin (BD Vacutainer, BD Biosciences, Franklin Lakes, NJ, USA), and plasma was separated by a centrifuge at a speed of 13,000 rpm for 10 min (4 °C). For lipid profile analysis of mice in all groups, total cholesterol (TC), glucose, triglyceride (TG), and high- and low-density lipoprotein cholesterol (HDL-C, LDL-C) were measured in plasma for lipid profile analysis. The arteriosclerosis index was calculated using these lipid profiles. Arteriosclerosis index analysis is as follows: TC/HDL-c, cardiac risk ratio 1; LDL-C/HDL-C, cardiac risk ratio 2; (TC-HDL-C)/HDL-C, atherogenic coefficient; TC-HDL-C ratio, atherogenic marker, and log (TG/HDL-c), atherogenic index were determined using the Ikewuchi equation [24,25,26]. Aspartate aminotransferase (AST) and alanine aminotransferase (ALT) were analyzed for hepatic function evaluation. All hematological analyses were performed using a FUJI DRI-CHEM animal clinical chemistry analyzer (NX700, FUJIFILM Corporation, Tokyo, Japan). Levels of lactate dehydrogenase (LDH), creatine kinase MB isoenzyme (CKMB), and creatine phosphokinase (CPK), which are cardiac injury biomarkers in plasma, were also measured using the identical chemistry analyzer. The plasma and liver tissue triglyceride levels (Cayman Chemical, Ann Arbor, MI, USA) were determined using commercially available ELISA kits.

### 2.7. Histological Analysis

The separated thoracic aorta and left ventricle tissues were fixed in 10% NBF (neutral buffered formalin, HT501128, Merk, Darmstadt, Hessen, Germany) solution for 24 h. The paraffin-embedded paraffin blocks were sliced with a microtome to a thickness of 6–7 μm, and the sliced tissue was attached to the slide (Thermo, Pittsburg, PA, USA). The thoracic aorta and left ventricle tissue slides were stained with Masson’s trichrome stain kit (BBC Biochemical, Mount Vernon, WA, USA), picrosirius red stain kit (24901, Polysciences, Warrington, PA, USA), and Orcein staining kit (ab245881, Abcam, Cambridge, Cambs, UK). In addition, liver and epididymal adipose tissue were fixed in 4% paraformaldehyde, and the tissue was cut using a frozen microtome (CM1900 Freezing Microtome, Leica, Wetzlar, Germany). The liver and epididymal fat tissue slides were stained with Oil Red O stain kit (ab150678, Abcam, Cambridge, Cambs, UK). The stained tissue was imaged with a microscope slide scanner (Motic Easy Scan Pro 1, National Optical and Scientific Instruments, Schertz, TX, USA) for histopathological analysis.

### 2.8. Immunohistochemistry (IHC) Staining

Cardiac tissue slides were immunostained by IHC staining kit (ab6464, Abcam, Cambridge, Cambs, UK). Tissue slides were incubated with primary antibodies of TGF-β1 and collagen IV (Santa Cruz Biotechnology, Santa Cruz, CA, USA). Histopathological comparisons were performed by Motic Easy Scan Pro 1 (microscope slide scanner, National Optical & Scientific Instruments, Inc., Schertz, TX, USA).

### 2.9. Western Blot Analysis and Antibodies

Left ventricular and renal tissues (30–40 μg of protein) were separated by electrophoresis (10% SDS-PAGE) and transferred to membranes. Then, they were washed with TBS-T (Tris-HCl, 10 mM; Tween-20, 0.05%; and NaCl, 150 mM) and blotted with 5% BSA (with 1X TBS-T) for 2 h. After another wash, incubation with appropriate primary antibodies (TGF-β, IL-1β, IL-6, and TNF-α) was performed overnight at 4 °C. Primary antibody tumor necrosis factor alpha (TNF-α, SC-1348), interleukin-6 (IL-6, SC-28343), interleukin-1β (IL-1β, SC-12742), and α-tubulin (SC-5286) were purchased from Santa Cruz (Santa Cruz Biotechnology, Dallas, TX, USA). The membranes were then washed with TBS-T and incubated with secondary antibodies for 2 h at room temperature. Protein expression levels were imaged with an iBright image analyzer (FL100, Thermo Fisher, Waltham, MA, USA). Image J was used to analyze protein expression levels (National Institutes of Health (NIH), Bethesda, MD, USA).

### 2.10. Statistical Analyses

Statistically significant differences between group means were determined by ANOVA and Bonferroni multiple comparison test or paired *t*-test. All experiments were repeated at least three to five times, and statistical analysis was performed by SigmaPlot (version 10) and GraphPad Prism (version 5). *p* < 0.05 was considered a statistically significant difference, and data are expressed as mean ± standard error.

## 3. Results

### 3.1. HPLC Chromatograms of Quercetin-3-O-Glucuronide from Blackcurrant

In the case of berries, raw fruits are sufficient, so heat treatment is rare. Heat treatment is a known process in food preservation that prevents and slows spoilage [27]. Therefore, blackcurrant has the disadvantage of being easily fermented due to its high sugar content, so heat treatment was performed on blackcurrant in consideration of various experimental methods. As a result of component analysis, active ingredients such as cyanidin-3-*O*-rutinoside and delphinidin-3-*O*-rutinoside (Figure 1A–D), which are known to have antioxidant effects despite heat treatment, were detected [28]. Among the anthocyanin standards, the contents of the four compounds known to be contained in blackcurrant extract were confirmed as delphinidin-3-*O*-glucoside (0.038 mg/g), delphinidin-3-*O*-rutinoside (0.374 mg/g), cyanidin-3-*O*-glucoside (0.314 mg/g), and cyanidin-3-*O*-rutinoside (0.423 mg/g) (Figure 1E).

### 3.2. Effect of Blackcurrant on Obesity and Fat Improvement in Type 2 Diabetic Mice

As shown in the image in Figure 2A, the T2DM mice had a larger body size than the control group (db/m mice, Aa), whereas the body size of the Bla.C treatment group was noticeably smaller than that of the T2DM group. Body weight values of T2DM mice for 8 weeks were significantly increased compared to control mice, whereas treatment with Bla.C or Vil.G significantly reduced the body weight of T2DM mice (Figure 2(Aa)). Additionally, Bla.C significantly reduced total epididymal fat pad weight, fat mass index (epididymal fat pad/body weight ratio, Figure 2(Bb)), adipose cell area (Figure 2(Bc)), and adipose cell diameter (Figure 2(Bd)) in T2DM mice. Therefore, it is considered that the weight loss of T2DM mice due to Bla.C treatment significantly reduced body weight by reducing total fat accumulation.

### 3.3. Blackcurrant Ameliorates Glucose Tolerance and Insulin Resistance in Type 2 Diabetes Mellitus Mice

To determine the effect of Bla.C on insulin resistance and glucose metabolism in T2DM mice, glucose tolerance, insulin levels, and fasting glucose levels were analyzed in plasma. As shown in Figure 3A, plasma insulin levels (a), insulin/glucose ratio (b), and HOMA-IR (c) indices were significantly increased in T2DM mice, whereas they were significantly decreased by Bla.C treatment. As shown in Figure 3B, the oral glucose tolerance test (OGTT) of Bla.C-treated T2DM mice was better than that of the T2DM group (a, b). In addition, it was confirmed that the HbA1c (c) of the Bla.C-treated mouse group showed a significant reduction effect when compared to the T2DM group, and the progression of hyperglycemia was alleviated (Figure 3B). These results suggest that Bla.C has a protective effect on glucose tolerance and insulin resistance in T2DM mice.

### 3.4. Blackcurrant Ameliorates Lipid Accumulation in the Liver

As a result of observing the liver by the naked eye, it was confirmed that the T2DM mouse had a larger liver, a brighter liver color, and more fat granules on the surface of the liver than the control group. On the other hand, it was confirmed that liver morphology was improved and surface fat granules were decreased in the Bla.C treatment group compared to the T2DM group (Figure 4A). Oil Red O (Figure 4B) and Orcein (Figure 4C) staining was performed to confirm the effect of Bla.C on hepatic steatosis and fibrosis. T2DM mice showed a significant increase in TG content in the liver (Figure 4(Ac)) and plasma (Figure 4(Da)) compared to controls, but levels were significantly decreased in the group treated with Bla.C. In addition, in the T2DM group, Bla.C treatment reduced elevated levels of ALT and AST, which are important blood biomarkers for liver function (Figure 4(Db,Dc)). Therefore, our findings confirmed that Bla.C reduced lipid accumulation in the liver of T2DM mice.

### 3.5. Blackcurrant Ameliorates Lipid Profile and Atherogenic Index Status in Type 2 Diabetes Mellitus Mice

Various studies have shown that the lipid profile is a major risk factor for atherosclerotic vascular and heart disease [29]. Therefore, reductions in lipid profile and atherogenesis index were considered to reduce the risk of cardiac disease. In T2DM mice, we found significant increases in plasma levels of TG, TC, and HDL-C; cardiac risk ratios; atherogenic coefficient; atherogenic marker; and atherogenic index. On the other hand, pretreatment with Bla.C inhibited increases in plasma lipid profile and atherogenic index (Table 1). These results suggest that Bla.C has protective effects on lipid metabolism and cardiovascular dysfunction in T2DM mice, which could partly explain the weight loss observed in T2DM mice.

### 3.6. Blackcurrant Alleviates Cardiac Dysfunction in Type 2 Diabetes Mellitus Mice

Echocardiography was performed in T2DM mice to determine whether Bla.C treatment had an improvement effect on cardiac function and wall thickness. Figure 5A is a representative M-mode image; echocardiographic data show that ejection fraction (EF) and fractional shortening (FS) were increased in T2DM mice, whereas they were significantly reduced with Bla.C treatment (Figure 5(Ba,Bb)). Left ventricular end-systolic volume (LVESV), left ventricular end-diastolic volume (LVEDV), left ventricular mass (LVd Mass), left ventricular internal dimension in diastole (LVIDd), and left ventricular posterior wall thickness (LVPWd) were significantly higher in T2DM mice compared to control, suggesting that cardiac function was impaired by diabetes. On the other hand, the cardiac dysfunction parameters were markedly reduced in Bla.C-treated mice compared to T2DM mice (Table 2).

In addition, it could be confirmed that the levels of plasma cardiac biomarkers lactate dehydrogenase (LDH, Figure 5(Ca)), creatine kinase MB isoenzyme (CKMB, Figure 5(Ca,Cb)), and creatine phosphokinase (CPK, Figure 5(Cc)) were increased in T2MD mice, but improved by Bla.C treatment. These results suggest that cardiac dysfunction occurs when diabetes occurs, and it can be confirmed that Bla.C has an improvement effect.

### 3.7. Effect of Blackcurrant on Cardiac Hypertrophy and Cardiovascular Fibrosis in Type 2 Diabetes Mellitus Mice

Histological analysis was performed to determine the protective effect of Bla.C against cardiac fibrosis in T2DM mice (Figure 6). Representative microscopic photographs in the cardiac tissue were stained by Masson’s trichrome staining (collagen fibers are stained blue color, Figure 6B) and Orcein staining (elastin fibers are stained bright red color, Figure 6C).

We confirmed the expression levels of the cardiac remodeling markers TGF-β1 (Figure 7A) and collagen IV (Figure 7B) by examining immunohistochemistry (IHC, expression of factors is shown in dark brown). TGF-β1 and collagen IV expression through IHC staining in the left ventricle tissues was increased in the T2DM group compared to the control group, whereas it was confirmed that it was decreased by Bla.C treatment.

### 3.8. Effect of Blackcurrant on Vascular Fibrosis and Inflammation

To confirm vascular fibrosis, elastin fibers were stained bright red with Orcein staining (Figure 7A) The thoracic aorta of T2DM mice had increased inner diameter (cross-sectional area of aorta, Figure 8(Ba)) and intimal thickness (length of tunica intima–media, Figure 8(Bb)) compared to controls. Protein levels of inflammatory cytokines (IL-6, interleukin-6; TNF-α, tumor necrosis factor alpha; IL-1β, interleukin-1β) were increased in T2DM mice, whereas Bla.C- or Vil.G-induced treatment markedly reduced the protein levels of T2DM mice (Figure 8(Ca–Cd)).

Therefore, Figure 9 is a representative image of the mechanism of amelioration of diabetic cardiovascular dysfunction by Bla.C treatment (Figure 9). Taken together, these results showed that treatment with Bla.C in T2DM mice could ameliorate diabetes cardiovascular complications by attenuating inflammatory cytokine activation signals.

## 4. Discussion

Type 2 diabetes mellitus (T2DM) is accompanied by related diseases such as obesity, high blood pressure, and dyslipidemia and increases the risk of heart failure and cardiac dysfunction such as coronary artery disease and hypertension [30]. This study demonstrated that Bla.C ameliorated the development of diabetes-induced cardiovascular complications by inhibiting cardiovascular fibrosis and inflammation on T2DM in db/db mice.

The T2DM mouse model, the db/db mouse, is a representative animal model of type 2 diabetes first reported in 1966 [31]. The T2DM mouse is currently used extensively in diabetic cardiomyopathy research because it recapitulates aspects of the metabolic syndrome associated with human obesity and resembles commonly used genetically modified models [32]. db/db mice have spontaneously mutated leptin receptors [33].

Thus, extreme leptin resistance in db/db mice leads to bulimia and consequent morbid obesity, as well as reproductive dysfunction and severe insulin resistance [34]. Therefore, db/db mice suffer from various metabolic diseases such as obesity, diabetes, and hypertension. Therefore, in this study, we investigated whether Bla.C extract could ameliorate diabetes-induced cardiovascular complications in T2DM mice.

The oral glucose tolerance test is used to screen for type 2 diabetes by measuring glucose tolerance. Continuous venous blood samples are collected from mouse tails to assess glucose and insulin, especially if the goal is to investigate the insulin secretion response [35]. Therefore, we administered 2 mg/kg of glucose by gavage to test glucose tolerance. Then, serial venous blood samples were collected to measure glucose levels. As a result, fasting insulin and glucose levels were increased in T2DM mice but decreased with Bla.C treatment. It was also confirmed that the oral glucose tolerance test results were improved in T2DM mice treated with Bla.C. However, the OGTT results have limitations in being considered as evidence that mice have increased insulin resistance. Moreover, the amount of glucose provided for OGTT should have been normalized by the differences in visceral adipose tissue [36]. Thus, one of the ways insulin removes glucose from the blood is by absorbing glucose from muscle and fat cells. Therefore, our results suggest only some of the possibilities of improving insulin resistance.

Several studies have shown that dyslipidemia risk factors total cholesterol, triglycerides, LDL-C, and LDL-C/HDL-C ratio are significantly greater in patients with heart disease than in healthy people [37]. In particular, HDL-C concentration was found to be inversely proportional to the risk of heart disease [38]. Therefore, the heart risk ratio, the arteriosclerosis coefficient, and the arteriosclerosis index are strong indicators of the risk of cardiovascular disease, and the higher the value, the higher the risk of cardiac disease [39].

The results of our study are consistent with those of other studies showing that the lipid profile, atherogenesis index, and cardiac risk ratio are greater in type 2 diabetic mice. Therefore, the reduction in lipid profile and atherosclerosis index due to blackcurrant treatment is thought to reduce the risk of heart disease. According to research, the HOMA-IR index divided by the product of insulin and glucose concentrations by a factor is a widely accepted score for analyzing type 2 diabetes [40]. In addition, hemoglobin A1c (HbA1c) in the blood can be considered as a biomarker for hyperglycemia suggestive of diabetes or prediabetes, and thus it has been utilized in research as a vascular and tissue biomarker of diabetes. It is a test that can accurately evaluate long-term blood glucose management as an excellent marker of metabolic health [41]. Our results showed that HOMA-IR index and HbA1c levels were significantly reduced in Bla.C-treated T2DM mice compared to T2DM mice. This Bla.C treatment suggests that plasma insulin and HOMA-IR reduction and improved glucose tolerance may improve insulin resistance in diabetes. Therefore, Bla.C is thought to prevent diabetic cardiovascular damage by improving insulin resistance.

Diabetes mellitus caused by impaired glycemic control and insulin resistance is a major risk factor for cardiovascular disease and is known to have a detrimental effect on the systolic function of the left ventricle. Diabetes also affects the heart muscle, causing systolic and diastolic heart failure [42]. Initially, the thickness of the left ventricle wall increases, the left ventricular compliance decreases, and the left ventricle volume gradually increases, causing left ventricular malformations [43]. From our results, it was confirmed that the left ventricular dysfunction caused by T2DM was improved by Bla.C treatment. In addition, various biomarkers, including creatine kinase MB isoenzyme (CK-MB), lactate dehydrogenase (LDH), and creatine phosphokinase (CPK), are considered hallmarks of cardiomyocytes, and their release into serum suggests death or dysfunction of these cardiac cells [44]. Our results were similarly consistent with increases in plasma LDH, CK-MB, and CPK activity in T2DM mice [45]. Similarly, in our results, left ventricular function and biomarkers deteriorated in db/db mice but were improved by Bla.C treatment, so Bla.C is considered to be effective in improving diabetic heart dysfunction.

Inflammation is known to play an important role in the pathogenesis of T2DM mice. The proinflammatory cytokines produced also have a direct effect on metabolic processes [46]. For example, TNF-α can ultimately lead to insulin resistance, reducing insulin sensitivity [47,48]. In addition, persistent hyperglycemia due to diabetes induces IL-6- and IL-1β-mediated dysfunction of insulin secretion [49]. In our study, we confirmed that the levels of inflammatory cytokines were significantly increased in T2DM mice in diabetes-induced cardiovascular disease. On the other hand, Bla.C treatment showed a relative decrease in the levels of inflammatory cytokines and improved the pathological tissue structure. TGF-β1 is a multifunctional cytokine that stimulates transcription factors such as collagen and fibronectin and contributes to various biological processes such as apoptosis, cell proliferation, and differentiation [50]. In addition, hyperglycemia plays an essential role in the pathogenesis of diabetic cardiovascular dysfunction by regulating the TGF-β signaling pathway [51]. Increased TGF-β1 signaling is upregulated in the impaired cardiovascular system and is involved in the pathogenesis of diabetic cardiopathy, leading to cardiac fibrosis and ventricular hypertrophy [52]. In addition, diabetic cardiopathy in T2DM mice was reported to increase the amount of collagen IV [53]. Our results showed that the expression of TGF-β1 and collagen IV was decreased in the TM2D mice group by treatment with Bla.C. This suggests that Bla.C alleviates cardiac fibrosis by downregulating the TGF-β signaling pathway. Therefore, our findings suggested that Bla.C may have significant protective effects against diabetic cardiovascular dysfunction. A recently published review suggested that nutraceuticals that improve markers of inflammation have the potential to play an adjuvant role in reducing the risk of cardiovascular disease [14]. Therefore, it is thought that the Bla.C extract used in this study has significance as a functional food because it inhibits inflammatory mediators.

However, our study acknowledges major limitations and requires further research. First, our results suggest only a fraction of the potential of Bla.C treatment to improve insulin resistance in type 2 diabetes. We regret not being able to measure insulin and glucose after normal eating. In addition, it would be nice to perform a lipid tolerance test to better define the lipid management effect of Bla.C at the liver level. Therefore, in future studies, confirmation through the insulin tolerance test (ITT) and lipid tolerance test (LTT) is necessary in order to reach a more definitive conclusion. Second, we must be clear about the purpose of using a boiling water extract rather than the fresh fruit of Bla.C. Therefore, further study is needed to compare and analyze the effects of Bla.C extract and fresh fruit.

## 5. Conclusions

The Bla.C extract improved diabetes-related metabolic disorders such as cardiovascular dysfunction and insulin resistance in T2DM mice. It also ameliorated heart inflammation and cardiovascular fibrosis. These results confirmed that Bla.C extract could play an essential role in preventing cardiovascular fibrosis, inflammation, and consequent diabetic heart disease, directly or indirectly, by improving blood sugar in diabetic conditions. Therefore, it is judged that the protective role of Bla.C extract against diabetes-related cardiovascular dysfunction can provide new insights into the development of functional foods and therapeutic agents for diabetes cardiovascular dysfunction.

## Figures and Tables

**Figure 1 nutrients-13-04177-f001:**
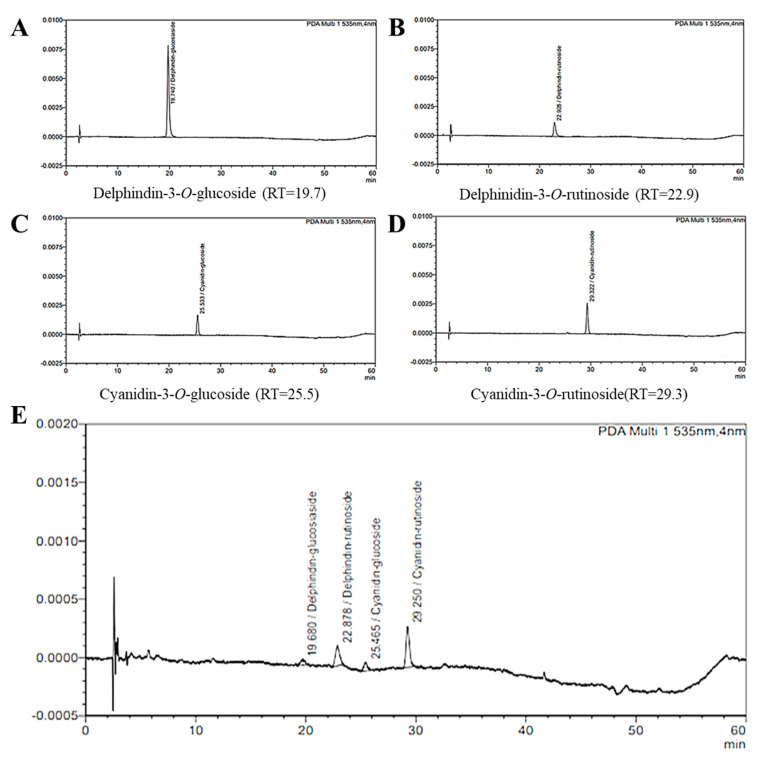
High−performance liquid chromatography (HPLC) chromatograms of quercetin-3-*O*-glucuronide from blackcurrant extract (detection at 330 nm). Standard compounds: delphindin-3-*O*-glucoside (RT = 19.7) (**A**); delphinidin-3-*O*-rutinoside (RT = 22.9) (**B**); cyanidin-3-*O*-glucoside (RT = 25.5) (**C**); cyanidin-3-*O*-rutinoside (RT = 29.3) (**D**). Blackcurrant extract (**E**).

**Figure 2 nutrients-13-04177-f002:**
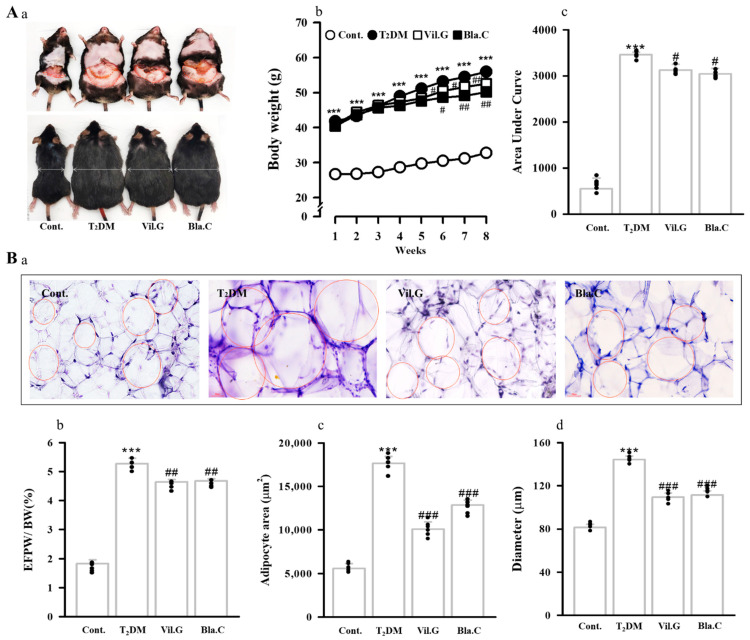
Effect of blackcurrant treatment on the changes in body weight and epididymal fat pad in type 2 diabetes mellitus mice (T_2_DM mice). Representative image of visceral fat distribution for each mice group (**A****a**). Changes in body weight ((**Ab**), *n* = 8 per group). Integral area under the curve (AUC, (**Ac**)). Representative microscope images of Oil Red O staining of epididymal fat pad ((**Ba**), *n* = 3 per group). Bottom panel indicates the weight, area, and size (**Bb**–**Bd**) of adipose cells. BW, body weight; EFPW, epididymal fat pad weight; Cont, control (db/m mice); T2DM, type 2 diabetes mellitus mice (db/db mice); Vil.G, db/db + vildagliptin 50 mg/kg/day, positive control, dipeptidyl peptidase-4 (DPP-4) inhibitor; Bla.C, db/db + blackcurrant 200 mg/kg/day. Data are shown as mean ± S.E. *** *p* < 0.001 vs. Cont.; # *p* < 0.05, ## *p* < 0.01, ### *p* < 0.001 vs. T_2_DM.

**Figure 3 nutrients-13-04177-f003:**
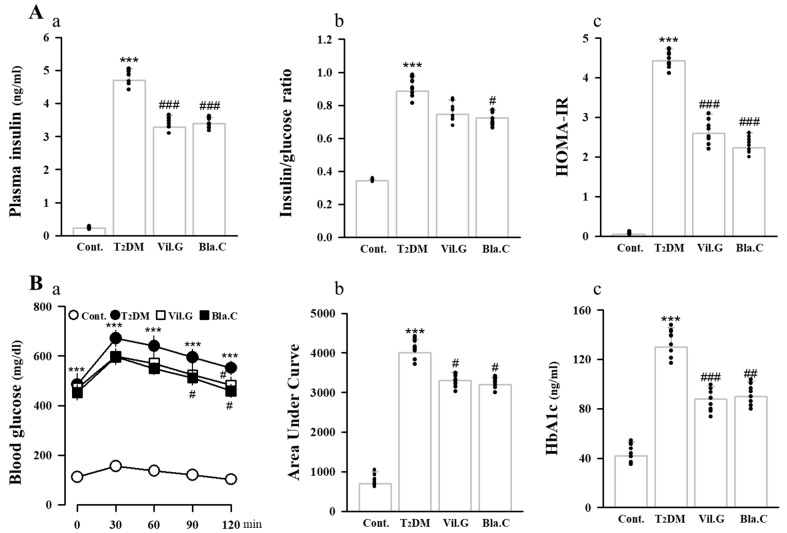
Hematological changes upon blackcurrant treatment in type 2 diabetes mellitus mice (T2DM mice). Measurements of insulin (**Aa**), insulin/glucose ratio (**Ab**), and HOMA-IR (**Ac**) in plasma. Oral administration of 2 mg/kg glucose for glucose tolerance test (OGTT, (**Ba**)). Glucose tolerance test expressed as area under curve (**Bb**). Hemoglobin A1c (HbA1c) test for diabetes (**Bc**). OGTT, oral glucose tolerance test; HbA1c, hemoglobin A1c or glycated hemoglobin; HOMA-IR, homeostatic model assessment for insulin resistance. Cont, control (db/m mice); T2DM, type 2 diabetes mellitus mice (db/db mice); Vil.G, db/db + vildagliptin 50 mg/kg/day, positive control, dipeptidyl peptidase-4 (DPP-4) inhibitor; Bla.C, db/db + blackcurrant 200 mg/kg/day. Data are shown as mean ± S.E. (*n* = 8 per group) *** *p* < 0.001 vs. Cont.; # *p* < 0.05, ## *p* < 0.01, ### *p* < 0.001 vs. T2DM.

**Figure 4 nutrients-13-04177-f004:**
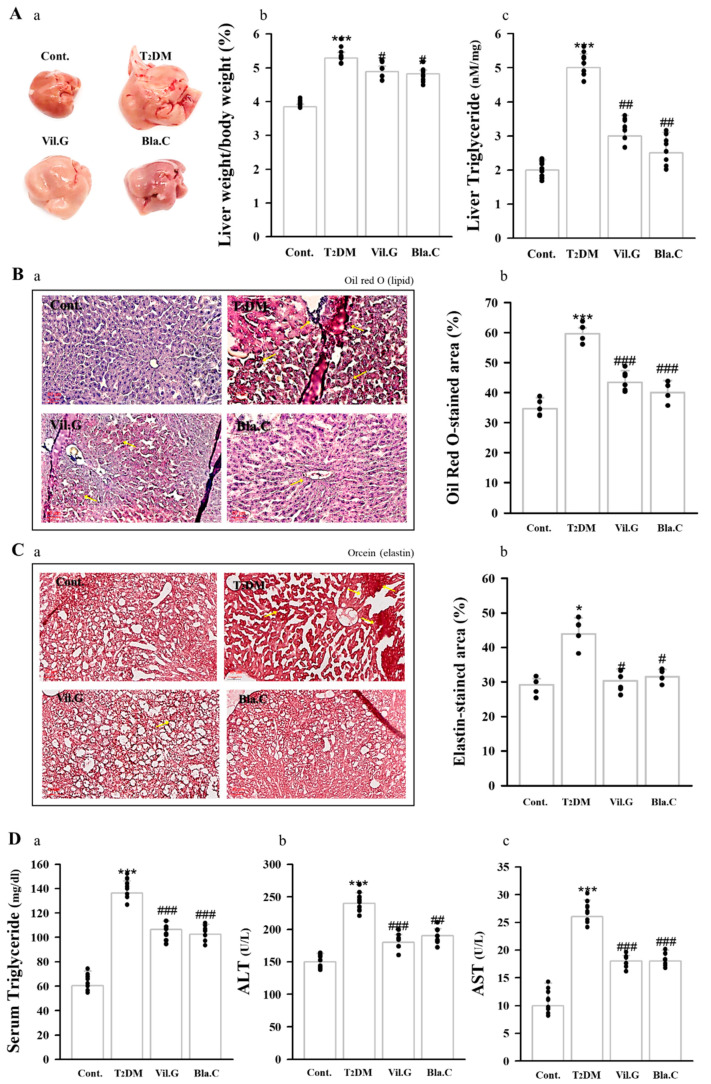
Effects of blackcurrant on hepatic histopathology and function in type 2 diabetes mellitus mice (T2DM mice). A representative photograph of the liver observed with the naked eye (**Aa**). Liver weight/body weight ratio (**Ab**) and liver triglyceride levels (**Ac**) of type 2 diabetic mice. Representative images of liver tissue stained with Oil Red O (**B**) and Orcein stain (**C**) to confirm histopathological changes (*n* = 3 per group). Yellow arrows indicate the location of lipids (**Ba**) and elastic fibers (**Ca**). Graphs quantifying the lipid-stained area (**Bb**) and elastin-stained area (**Cb**). Graphic of serum triglyceride (**Da**), ALT (**Db**), and AST (**Dc**) plasma levels in db/db mice (*n* = 8 per group) Cont, control (db/m mice); T2DM, type 2 diabetes mellitus mice; Vil.G, T2DM + vildagliptin 50 mg/kg/day, positive control, dipeptidyl peptidase-4 (DPP-4) inhibitor; Bla.C, T2DM + blackcurrant 200 mg/kg/day; AST, aspartate aminotransferase; ALT, alanine aminotransferase. Data are shown as mean ± S.E. * *p* < 0.05, *** *p* < 0.001 vs. Cont.; # *p* < 0.05, ## *p* < 0.01, ### *p* < 0.001 vs. T2DM.

**Figure 5 nutrients-13-04177-f005:**
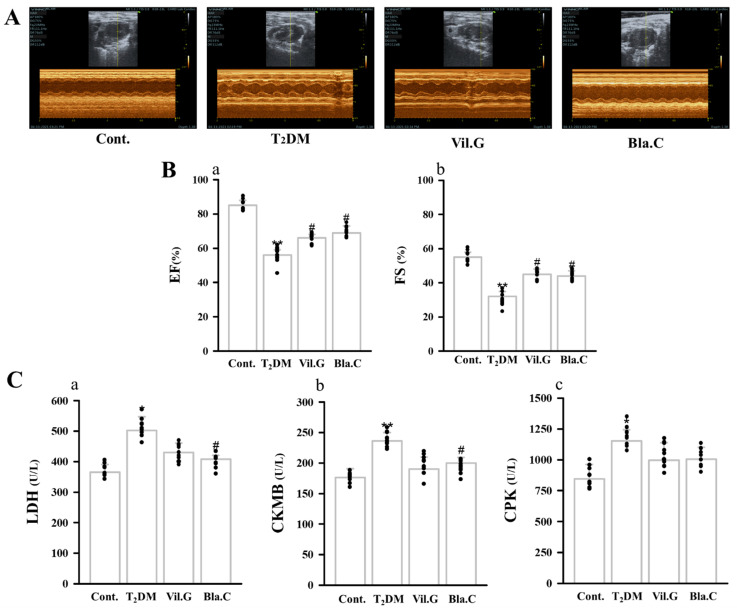
Effect of blackcurrant on left ventricle (LV) remodeling in type 2 diabetes mellitus mice (T2DM mice). Representative echocardiography images (M-mode) for each group (**A**). Changes of cardiac function of EF (**Ba**) and FS (**Bb**) were measured echocardiography in each group (*n* = 8 per group). Changes in plasma of the cardiac biomarkers LDH (**Ca**), CKMB (**Cb**), and CPK (**Cc**) (*n* = 8 per group). EF, ejection fraction; FS, fractional shortening; LDH, lactate dehydrogenase; CKMB, creatine kinase MB isoenzyme; CPK, creatine phosphokinase; Cont, control (db/m mice); T2DM, type 2 diabetes mellitus mice (db/db mice); Vil.G, db/db + vildagliptin 50 mg/kg/day, positive control, dipeptidyl peptidase-4 (DPP-4) inhibitor; Bla.C, db/db + blackcurrant 200 mg/kg/day. Data are shown as mean ± S.E. * *p* < 0.05 and ** *p* < 0.01 vs. Cont.; # *p* < 0.05 vs. T2DM.

**Figure 6 nutrients-13-04177-f006:**
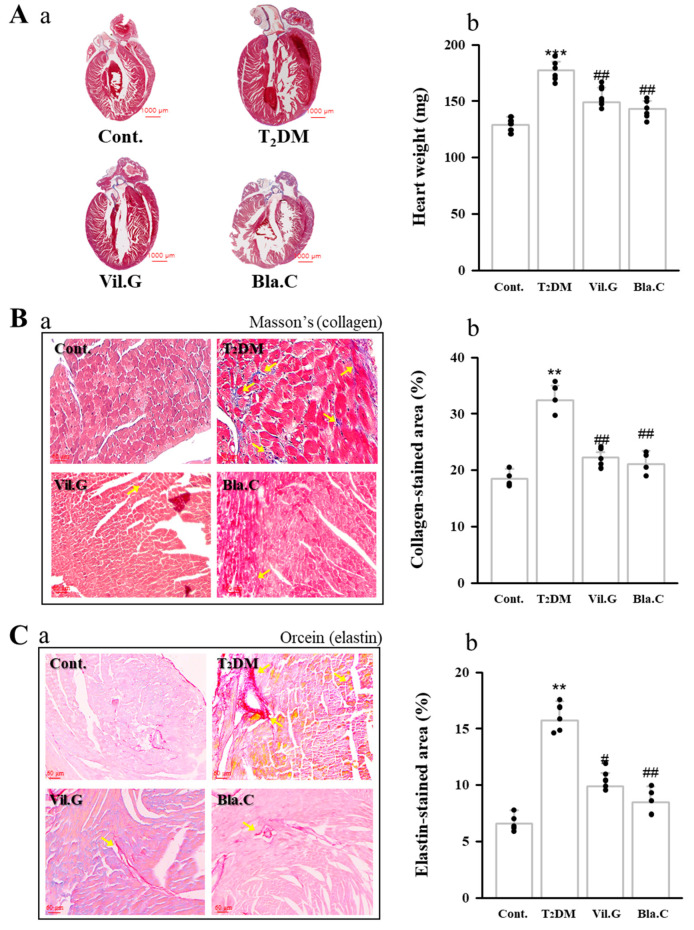
Effects of blackcurrant on diabetic cardiomyopathy in type 2 diabetes mellitus mice (T2DM mice). Blackcurrant attenuates hypertrophy (**A**) and fibrosis (**B**,**C**) of myocytes in T2DM mice. Full size image (**Aa**) weight graph (**Ab**) the heart to confirm cardiac hypertrophy. Representative images of collagen with Masson’s trichrome stain (**Ba**) and elastic fibers with Orcein stain (**Ca**). Yellow arrows indicate the location of fibrosis of collagen fibers (**B**) and elastic fibers (**C**) in the left ventricle. Graphs quantifying the collagen-stained area (**Bb**) and elastin-stained area (**Cb**). Cont, control (db/m mice); T2DM, type 2 diabetes mellitus mice (db/db mice); Vil.G, db/db+vildagliptin 50 mg/kg/day, positive control, dipeptidyl peptidase-4 (DPP-4) inhibitor; Bla.C, db/db+blackcurrant 200 mg/kg/day. Data are shown as mean ± S.E. ** *p* < 0.01 and *** *p* < 0.001 vs. Cont.; # *p* < 0.05 and ## *p* < 0.01 vs. T2DM.

**Figure 7 nutrients-13-04177-f007:**
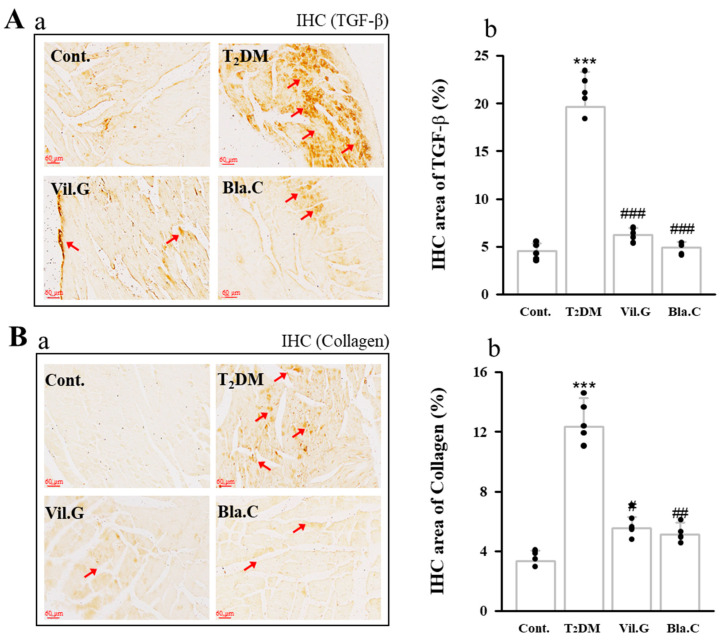
Effects of blackcurrant on diabetic cardiomyopathy in type 2 diabetes mellitus mice (T2DM mice). The expression levels of TGF-β (**Aa**) and collagen (**Ba**) were confirmed in T2DM mice cardiac tissue slides using immunohistochemistry (IHC) staining in T2DM (magnification 400×, *n* = 3 per group). Red arrows indicate the location of TGF-β (D) and collagen IV (D) expressed by IHC in left ventricle. Graphs quantifying the IHC-stained expression area of TGF-β (**Ab**) and IHC-stained expression area of collagen (**Bb**). Cont, control (db/m mice); T2DM, type 2 diabetes mellitus mice (db/db mice); Vil.G, db/db + vildagliptin 50 mg/kg/day, positive control, dipeptidyl peptidase-4 (DPP-4) inhibitor; Bla.C, db/db + blackcurrant 200 mg/kg/day; TGF-β, transforming growth factor-β. Data are shown as mean ± S.E. *** *p* < 0.001 vs. Cont.; # *p* < 0.05, ## *p* < 0.01, ### *p* < 0.001 vs. T2DM.

**Figure 8 nutrients-13-04177-f008:**
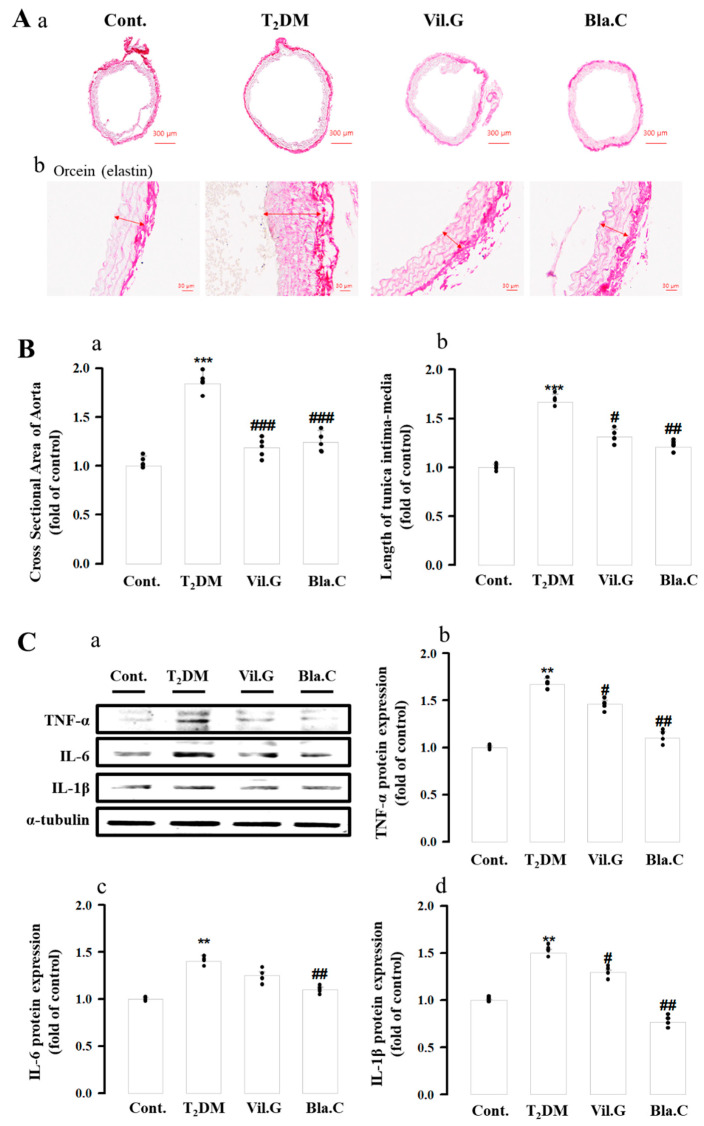
Effects of blackcurrant on diabetic cardiomyopathy in type 2 diabetes mellitus mice (T2DM mice). Blackcurrant attenuates vascular inflammation and fibrosis of thoracic aorta in T2DM mice. Representative images of Orcein staining in T2DM thoracic aortic tissue ((**Aa**), magnification × 40; (**Ab**), magnification × 400, *n* = 4 per group). Graph of the effect of Bla.C on the change in cross-sectional area (**Ba**) and length of tunica intima–media (**Bb**) of the thoracic aorta in T2MD mice. The expression of inflammatory cytokines in the thoracic aorta was determined by Western blot analysis (**C****a**–**C****d**) (*n* = 4 per group). Cont, control (db/m mice); T2DM, type 2 diabetes mellitus mice (db/db mice); Vil.G, db/db + vildagliptin 50 mg/kg/day, positive control, dipeptidyl peptidase-4 (DPP-4) inhibitor; Bla.C, db/db + blackcurrant 200 mg/kg/day; TNF-α, tumor necrosis factor alpha; IL-6, interleukin-6; IL-1β, interleukin-1β. Data are shown as mean ± S.E. ** *p* < 0.01, *** *p* < 0.001 vs. Cont.; # *p* < 0.05, ## *p* < 0.01, ### *p* < 0.001 vs. T2DM.

**Figure 9 nutrients-13-04177-f009:**
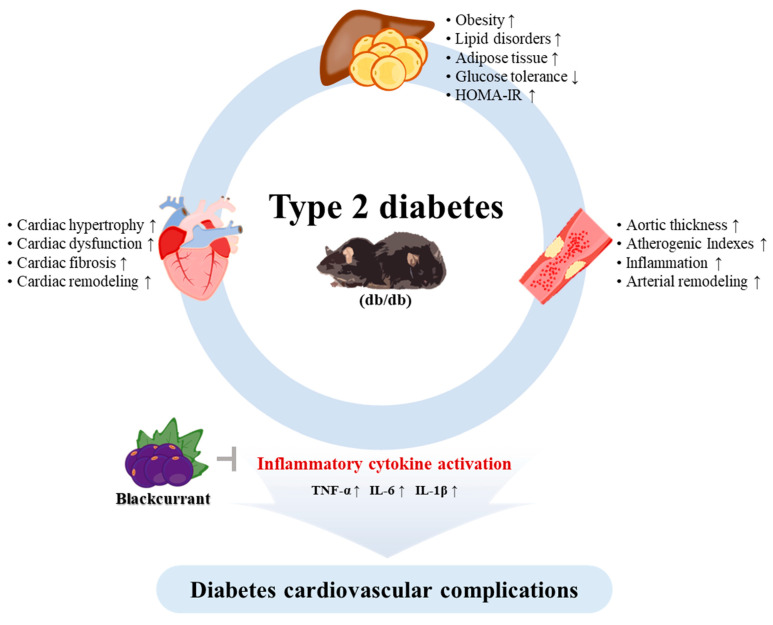
Schematic diagram of the effect of blackcurrant in improving diabetes cardiovascular complications. db/db, type 2 diabetes mellitus mice; TNF-α, tumor necrosis factor alpha; IL-6, interleukin-6; IL-1β, interleukin-1β; HOMA-IR, homeostatic model assessment for insulin resistance; ↑, increase; ↓, decrease.

**Table 1 nutrients-13-04177-t001:** Effect of blackcurrant on plasma parameters in type 2 diabetes mellitus mice.

Parameters	Cont.			T2DM			Vil.G			Bla.C		
Cardiac risk ratio 1 (TC/HDL-C)	1.217	±	0.019	1.663	±	0.086 **	1.298	±	0.017 ^##^	1.347	±	0.080 ^##^
Cardiac risk ratio 2 (LDL-C/HDL-C)	1.600	±	0.054	2.504	±	0.246 **	1.651	±	0.095 ^##^	1.398	±	0.097 ^##^
Atherogenic coefficient (TC-HDL)/HDL-C	0.217	±	0.028	0.463	±	0.096 ***	0.298	±	0.024 ^###^	0.292	±	0.026 ^##^
Atherogenic marker (TG/HDL-C ratio)	0.885	±	0.095	1.285	±	0.182 ***	1.138	±	0.120 ^#^	1.038	±	0.238 ^#^
Atherogenic index log (TG/HDL-C)	0.330	±	0.046	0.475	±	0.064 **	0.459	±	0.040 ^#^	0.343	±	0.038 ^##^

Cont, control, db/m mice; T2DM, db/db mice, type 2 diabetes mellitus mice; Vil.G, db/db + vildagliptin 50 mg/kg/day, positive control, dipeptidyl peptidase-4 (DPP-4) inhibitor; Bla.C, db/db + blackcurrant 200 mg/kg/day; TC, total cholesterol; TG, triglyceride; HDL-C, high-density lipoprotein cholesterol; LDL-C, low-density lipoprotein cholesterol. Data are shown as mean ± S.E. (*n* = 8 per group). ** *p* < 0.01, *** *p* < 0.001 vs. Cont.; ^#^
*p* < 0.05, ^##^
*p* < 0.01, ^###^
*p* < 0.001 vs. T2DM.

**Table 2 nutrients-13-04177-t002:** Protective effect of blackcurrant on diabetes-induced cardiac dysfunction.

Characteristics	Cont.			T2DM			Vil.G			Bla.C		
LVEDV (mL)	0.098	±	0.013	0.185	±	0.038 ***	0.105	±	0.012 ^##^	0.121	±	0.011 ^##^
LVESV (mL)	0.013	±	0.002	0.072	±	0.020 ***	0.018	±	0.002 ^###^	0.032	±	0.009 ^###^
LVd Mass (g)	0.109	±	0.007	0.204	±	0.040 ***	0.106	±	0.008 ^#^	0.111	±	0.007 ^##^
LVIDd (mm)	3.328	±	0.134	4.017	±	0.266 **	3.395	±	0.126 ^##^	3.609	±	0.108 ^#^
LVPWd (mm)	1.038	±	0.107	1.253	±	0.107 **	0.893	±	0.071 ^##^	0.960	±	0.050 ^##^

Cont, control, db/m mice; T2DM, db/db mice, type 2 diabetes mellitus mice; Vil.G, db/db + vildagliptin 50 mg/kg/day, positive control, dipeptidyl peptidase-4 (DPP-4) inhibitor; Bla.C, db/db + blackcurrant 200 mg/kg/day; LVEDV, left ventricular end-diastolic volume; LVESV, left ventricular end-systolic volume; LVd Mass, left ventricular mass; LVIDd, left ventricular internal dimension in diastole; LVPWd, left ventricle posterior wall thickness. Data are shown as mean ± S.E. (*n* = 8 per group). ** *p* < 0.01, *** *p* < 0.001 vs. Cont; # *p* < 0.05, ## *p* < 0.01, ### *p* < 0.001 vs. T2DM.

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
