# Peer review of "Blackcurrant Improves Diabetic Cardiovascular Dysfunction by Reducing Inflammatory Cytokines in Type 2 Diabetes Mellitus Mice"

_nutrients, 2021, doi:10.3390/nu13114177_

Round 1
Reviewer 1 Report
Second review: While examining the earlier submission and comparing to the present form,there has been substantial improvements in the figures of the manuscript.Some of the results/graphs have been redone and are more accurate and presentable.
There are still some queries like in the method section as to why the extract was boiled at 100C instead at room temperature.There needs to be a clarification.Secondly, the authors need to mention the toxicitity profile if done previously by some other research group to certify the dosage used in these experiments.If the extract is not toxic, it needs to be stated.
The english language has been considerably improved.
Author Response
Reviewer 1
Second review: While examining the earlier submission and comparing to the present form,there has been substantial improvements in the figures of the manuscript. Some of the results/graphs have been redone and are more accurate and presentable.
I really thank you very much for your invaluable suggestions and comments.
There are still some queries like in the method section as to why the extract was boiled at 100C℃instead at room temperature. There needs to be a clarification.
→ In the case of berries, raw fruits are sufficient, so heat treatment is rare. Heat treatment is a known process in food preservation that prevents and slows spoilage. Therefore, black currant has a disadvantage of being easily fermented due to its high sugar content, so heat treatment was performed on blackcurrant in consideration of various experimental methods. As suggested by the reviewer, we have revised the content of blackcurrant heat-treating at 100 °C and added the reference to the manuscript (page 5, section 3.1).
(Added reference list)
- Korzenszky, P.; Molnár, E. Examination of heat treatments at preservation of grape must. Potravinarstvo. 2014, 8(1)1, 38-42.
- León-Gonzáleza, A.J.; Sharifa, T.; Kayali, A.; Abbas, M.; Dandache, I.; Etienne-Selloum, N.; Kevers, C.; Pincemail, J.; Auger. C., Chabert, P.; Alhosin, M.; Schini-Kertha, V.B. Delphinidin-3-O-glucoside and delphinidin-3-O-rutinoside mediate the redox-sensitive caspase 3-related pro-apoptotic effect of blackcurrant juice on leukaemia Jurkat cells. J Funct Foods. 2015, 17, 847-856
Secondly, the authors need to mention the toxicitity profile if done previously by some other research group to certify the dosage used in these experiments. If the extract is not toxic, it needs to be stated.
→ Because it was a commonly eaten berry, toxicity was not studied. However, since the concentration was established based on previous studies, content and reference were added to section 2.3 (page 3).
(Added reference list)
- Park et al. Blackcurrant Suppresses Metabolic Syndrome Induced by High-Fructose Diet in Rats. Evid. Based Complement. Alternat. Med. 2015, 385976.
The english language has been considerably improved.
→ Thanks for reviewing the manuscript submitted, and we really thank Reviewer #1 very much indeed.
Reviewer 2 Report
The conclusion should be smotheen. I do not believe it is correct to state that a nutraceutical can "have a significant effect on the prevention of cardiovascular fibrosis, inflammatory and consequent diabetes-induced cardiovascular complications". Please change as "ameliorate".
Author Response
Reviewer 2
The conclusion should be smotheen. I do not believe it is correct to state that a nutraceutical can "have a significant effect on the prevention of cardiovascular fibrosis, inflammatory and consequent diabetes-induced cardiovascular complications". Please change as "ameliorate".
→ I really thank you very much for your invaluable suggestions and comments. As suggested by the reviewer, "significant effect" was changed to "ameliorated ". Thanks for reviewing the manuscript submitted, and we really thank Reviewer #2 very much indeed.
This manuscript is a resubmission of an earlier submission. The following is a list of the peer review reports and author responses from that submission.
Round 1
Reviewer 1 Report
- Most of the results could be affected by a wrong application of statistical analysis. Students’ TTest is inappropriate (no Gaussian distribution has been demonstrated); instead one-way ANOVA followed by a post-test has to be performed. It is not clear how many animals have been used for each experiment.
- How many experiments have been performed for figure 7D? Instead of presenting WB analyses of cytokines, ELISA assays have to be performed.
- The Authors have to perform ITT (0.2 UI/L)
- Why did the Authors mention "subjects" in section 2.6? Did they perform experiments in humans?
- In which table the lipid profile is shown?
- Relative to section 2.3, which was the route of administration of drugs? Relative to blackcurrant, daily oral gavage for 10 weeks is quite challenging. Which was the inflammatory milieu of gastric mucosa? Why did the Authors choose the concentration of 200 mg/Kg/day in the case of blackcurrant?
- Introduction has to be rewrote highlighting the impact of nutraceuticals on inflammation (Prog Cardiovasc Dis. 2021 Jul-Aug;67:40-52.). Be focused on the main aim. Lane 49 indicates "inflammatory cytokine interleukine (IL)"; which interleukin did the Authors refer to?
- Figure legends have to be self explanatiry reproting also the numerosity of each experiment.
Reviewer 2 Report
#1.The introduction can be made more elaborate and related to the topic under consideration.
2. The authors mention in the Materials and Methods section about the preparation of the blackcurrant extract by boiling at 100C.Generally, as a rule the extracts are made by mixing the botanicals in solvent/water at room temp or at 37C with continous shaking.The 100C will render the bioactive compounds unsuitable.Please explain the use of high temperature.
3.The authors mention the dose provided to the mice under Experimental animals section as 200mg/kgbw/day.How was this dose determined?Was there any pilot study done or any MTD (Maximum Tolerance Dose) experiment done to determine the dosage.Were any side-effects or toxicity found in histopathological samples/tissues like liver,kidneys etc.
4 The figure 6 especially need attention where the tissue slides shown are of poor quality.The magnification bar is missing,the size of the pictures are too small.The authors may add arrows or marks to distinguish the area of interest to be shown.
#5.The western blot data shown in Figure 7 does not show proper bands as they are blurred and cannot be properly evaluated.
Round 2
Reviewer 1 Report
I suggest to show the standard deviation of each bar graph with dots. It will clarify definitively the numerosity of each sample. Relative to the administration of blackcurrent, since the Authors reported it was by water, I was wondering in which way the Authors are sure the daily dose assumed by each animal was the same?
Author Response
Thanks for reviewing the manuscript submitted, and we really thank Reviewer #2 very much indeed.
Comments 1: I suggest to show the standard deviation of each bar graph with dots. It will clarify definitively the numerosity of each sample.
Response 1: I really thank you very much for your invaluable suggestions and comments. We have revised the Figure 2, 3, 4, 5, and 7 as suggested.
Comments 2: Relative to the administration of blackcurrent, since the Authors reported it was by water, I was wondering in which way the Authors are sure the daily dose assumed by each animal was the same?
Response 2: Prior to the start of this experiment, oral intake (with zonde) using ICR mice was performed, but since the number of sacrificed ICR mice was large, this method was chosen to prevent unnecessary sacrifices from continuing. To ensure consistent drug intake, we measured water intake during the acclimatization period to determine the amount of drinking water per animal. We tried to set the amount of drinking water per animal, dissolve the drug according to the amount, and adjust the intake as much as possible. Of course, it is difficult to say that all mice took the same amount of blackcurrant, but in the case of mice, it is difficult to administer orally for a long time. Oral administration is very effective, but is technically difficult, especially in small animals such as mice, and is known to occasionally cause esophageal or other injuries. Therefore, several studies are being conducted to develop a method for spontaneous oral administration of substances to laboratory mice (1-3). Therefore, we conducted a study with reference to several studies in which mice were induced to take drugs voluntarily, in order to minimize sacrifice due to the lack of skills in handling mice (4-8).
- Zhang. Method for voluntary oral administration of drugs in mice. STAR Protoc. 2021 Feb 5;2(1):100330.
- Hovard et, al. The applicability of a gel delivery system for self-administration of buprenorphine to laboratory mice. Lab Anim. 2015 Jan;49(1):40-5.
- Zapata et al. Self-Administration of Drugs in Mouse Models of Feeding and Obesity. J Vis Exp . 2021 Jun 8;(172):10.3791/62775.
- Lee et al. Portulaca oleracea Ameliorates Diabetic Vascular Inflammation and Endothelial Dysfunction in db/db Mice. Evid Based Complement Alternat Med. 2012;2012:741824.
- Hwang et al. Anti-diabetic atherosclerosis effect of Prunella vulgaris in db/db mice with type 2 diabetes. Am J Chin Med . 2012;40(5):937-51.
- Yoon et al. Dianthus superbus Improves Glomerular Fibrosis and Renal Dysfunction in Diabetic Nephropathy Model. Nutrients. 2019;11(3): 553.
- Otsuki et al., Vascular endothelium as a target of beraprost sodium and fenofibrate for antiatherosclerotic therapy in type 2 diabetes mellitus. Vasc Health Risk Manag. 2005;1(3):209-15.
- Guo et al. Resveratrol ameliorates diabetic vascular inflammation and macrophage infiltration in db/db mice by inhibiting the NF-κB pathway. Diab Vasc Dis Res. 2014 Mar;11(2):92-102.